# Expanding Broad Molecular Reflex Testing in Non-Small Cell Lung Cancer to Squamous Histology

**DOI:** 10.3390/cancers16050903

**Published:** 2024-02-23

**Authors:** Martin Zacharias, Selma Konjic, Nikolaus Kratochwill, Gudrun Absenger, Angelika Terbuch, Philipp J. Jost, Robert Wurm, Jörg Lindenmann, Karl Kashofer, Franz Gollowitsch, Gregor Gorkiewicz, Luka Brcic

**Affiliations:** 1Diagnostic and Research Institute of Pathology, Medical University of Graz, 8010 Graz, Austria; martin.zacharias@medunigraz.at (M.Z.); selma.konjic@medunigraz.at (S.K.); nikolaus.kratochwill@medunigraz.at (N.K.); karl.kashofer@medunigraz.at (K.K.); franz.gollowitsch@medunigraz.at (F.G.); gregor.gorkiewicz@medunigraz.at (G.G.); 2Division of Oncology, Department of Internal Medicine, Medical University of Graz, 8010 Graz, Austria; gudrun.absenger@medunigraz.at (G.A.); angelika.terbuch@medunigraz.at (A.T.); philipp.jost@medunigraz.at (P.J.J.); 3Division of Pulmonology, Department of Internal Medicine, Medical University of Graz, 8010 Graz, Austria; robert.wurm@medunigraz.at; 4Division of Thoracic and Hyperbaric Surgery, Department of Surgery, Medical University of Graz, 8010 Graz, Austria; jo.lindenmann@medunigraz.at

**Keywords:** lung cancer, squamous cell carcinoma, molecular testing, next-generation sequencing

## Abstract

**Simple Summary:**

Targeted therapies have revolutionized the treatment of patients with non-small cell lung cancer (NSCLC). Adenocarcinoma, the most common histological subtype of NSCLC, is the posterchild of this therapeutic approach because of its high rate of treatable targets. In our study, we provide evidence that squamous cell carcinoma (SCC), the second most common histological subtype of NSCLC, also harbors a significant number of treatable targets, although at a lower rate than adenocarcinoma. Furthermore, we show that most SCC patients harboring such an alteration were >50 years of age and current or former smokers, questioning restrictive molecular testing strategies. Based on our data, we propose that broad molecular profiling should be performed in all newly diagnosed NSCLC cases, irrespective of histological subtype, but also irrespective of age or smoking status.

**Abstract:**

Due to the success story of biomarker-driven targeted therapy, most NSCLC guidelines agree that molecular reflex testing should be performed in all cases with non-squamous cell carcinoma (non-SCC). In contrast, testing recommendations for squamous cell carcinoma (SCC) vary considerably, specifically concerning the exclusion of patients of certain age or smoking status from molecular testing strategies. We performed a retrospective single-center study examining the value of molecular reflex testing in an unselected cohort of 316 consecutive lung SCC cases, tested by DNA- and RNA-based next-generation sequencing (NGS) at our academic institution between 2019 and 2023. Clinicopathological data from these cases were obtained from electronic medical records and correlated with sequencing results. In 21/316 (6.6%) cases, we detected an already established molecular target for an approved drug. Among these were seven cases with an *EGFR* mutation, seven with a *KRAS* G12C mutation, four with an *ALK* fusion, two with an *EGFR* fusion and one with a METex14 skipping event. All patients harboring a targetable alteration were >50 years of age and most of them had >15 pack-years, questioning restrictive molecular testing strategies. Based on our real-world data, we propose a reflex testing workflow using DNA- and RNA-based NGS that includes all newly diagnosed NSCLC cases, irrespective of histology, but also irrespective of age or smoking status.

## 1. Introduction

Targeted therapies have revolutionized the treatment of non-small cell lung carcinoma (NSCLC). However, treatment efficacy is largely dependent on the upregulation of specific oncogenic pathways in tumor cells and, therefore, the detection of the respective molecular alterations is a prerequisite before treatment initiation [1]. Molecular testing approaches for NSCLC have changed dramatically within the last years. Today, next-generation sequencing (NGS) is regarded as the gold standard method for clinically relevant molecular profiling, especially as the landscape of potentially targetable genetic alterations expands rapidly [2].

The two main entities comprising NSCLC are adenocarcinoma (AC) and squamous cell carcinoma (SCC) [3]. Since the introduction of targeted therapies, AC has been in the spotlight because of its relatively high proportion of targetable genetic alterations [4,5]. International guidelines for the management of NSCLC patients strongly recommend reflex broad molecular profiling of all newly diagnosed AC cases and mostly agree that *ALK*, *BRAF*, *ERBB2*, *KRAS*, *MET*, *NTRK*, *RET*, and *ROS1* must be included in testing panels [6,7,8].

In contrast to AC, molecular testing recommendations for SCC are more controversial. The latest molecular testing guidelines from the College of American Pathologists (CAP), the International Association for the Study of Lung Cancer (IASLC) and the Association for Molecular Pathology (AMP) recommend testing in non-AC only in patients with minimal (1–10 packs per years) or absent tobacco exposure and in patients younger than 50 years [7]. In line with this, the recently published European Society for Medical Oncology (ESMO) Clinical Practice Guideline for oncogene-addicted NSCLC does not recommend molecular testing in patients with a confident diagnosis of SCC, except unusual cases like never/former light smokers (≤15 pack-years) or young (<50 years) patients [6]. In contrast, the current National Comprehensive Cancer Network Clinical Practice Guidelines in Oncology (NCCN Guidelines) for NSCLC recommend that “molecular testing may be considered in all patients with metastatic NSCLC squamous cell carcinoma and not just those with certain characteristics, such as never smoking status and mixed histology.” [8].

At our institution (Medical University of Graz, Graz, Austria), pathologist-initiated reflex molecular testing has been routinely performed for several years in all newly diagnosed NSCLCs, independent of histology. This broad molecular profiling strategy enabled us to perform a single-center study examining the value of reflex testing in a large, unselected cohort of consecutive lung SCC cases. With this study, we add real-world evidence to the above-described controversy between different international NSCLC guidelines.

## 2. Materials and Methods

### 2.1. Patient Cohort

We included all newly diagnosed NSCLC cases with squamous histology tested by DNA- and RNA-based NGS at our academic institution (Diagnostic and Research Institute of Pathology, Medical University of Graz, Austria) between 1 January 2019 and 30 September 2023 (n = 316). Exclusion criteria were incomplete molecular testing and/or incomplete NGS data. Since our study cohort comprises all consecutive cases (irrespective of age, smoking status, and tumor stage), a selection bias can be excluded. Clinical, pathological, and molecular data were retrospectively obtained from electronic medical records. The study was conducted according to the Helsinki Declaration guidelines and was approved by the Ethics Committee of the Medical University of Graz (33-066 ex 20/21).

### 2.2. DNA-Based Next-Generation Sequencing

DNA-based next-generation sequencing was performed as previously described [9]. Briefly, DNA extraction was performed for between 2 and 12 FFPE tissue sections for each case using the Maxwell 16 instrument (Promega, Mannheim, Germany) and the Maxwell RSC DNA FFPE Kit (Promega, Mannheim, Germany; CatNr: AS1450). After quantification of DNA via picogreen fluorescence, library preparation was performed with 10 ng of DNA. The libraries were prepared using the AmpliSeq library kit 2.0 (Thermo Fisher Scientific, Waltham, MA, USA) and the Ion Ampliseq Colon and Lung Cancer Research Panel v2 primer pool covering hotspot mutations in 22 genes (*ALK*, *AKT1*, *BRAF*, *CTNNB1*, *DDR2*, *EGFR*, *ERBB2*, *ERBB4*, *FBXW7*, *FGFR1*, *FGFR2*, *FGFR3*, *KRAS*, *MAP2K1*, *MET*, *NOTCH1*, *NRAS*, *PIK3CA*, *PTEN*, *SMAD4*, *STK11*, *TP53*). Sequencing was performed on an Ion S5XL benchtop sequencer (Thermo Fisher Scientific, Waltham, MA, USA). The initial data analysis was performed using the Ion Torrent Suite Software Plug-ins (Thermo Fisher Scientific, open source, GPL, https://github.com/iontorrent/, accessed on 20 February 2024), and was performed as described previously [9]. In short, this included base calling, alignment to the HG19 reference genome via TMAP mapper, and variant calling via a modified diBayes approach. Called variants were annotated via the open-source software ANNOVAR ([10]) and SnpEff ([11]). As a next step, all coding and nonsynonymous mutations were further evaluated and visually inspected using IGV (http://www.broadinstitute.org/igv/, accessed on 20 February 2024). Variant calls resulting from sequence effects or technical read errors were excluded.

### 2.3. RNA-Based Next-Generation Sequencing

RNA-based next-generation sequencing was also performed as previously described [9]. Briefly, RNA extraction was performed on from five to eight FFPE tissue sections for each case using the Maxwell RSC RNA FFPE Kit (Promega, Mannheim, Germany). After quantification of RNA via ribogreen fluorescence (Qubit fluorometer, Life Tech Austria, Vienna, Austria), 250 ng of RNA was used for further analyses with the Archer FusionPlex Expanded Lung 18090 v1.0 primer pool (ArcherDX, Boulder, CO, USA). This panel included the genes *ALK*, *BRAF*, *EGFR*, *ERBB2*, *FGFR1*, *FGFR2*, *FGFR3*, *MET*, *NRG1*, *NTRK1*, *NTRK2*, *NTRK3*, *NUTM1*, *PIK3CA*, *RET*, and *ROS1*. Sequencing was performed on an Ion S5XL benchtop sequencer (Thermo Fisher Scientific, Waltham, MA, USA) and Ion 550 chip kit. Sequencing data were analyzed with the ArcherDX Analysis software (ArcherDX, Boulder, CO, USA). Translocations called with more than 10 individual reads were included in the final results.

### 2.4. Validation of Assays for DNA and RNA-Based Analyzes

A thorough validation of all in-house NGS assays was performed as previously described [9]. In short, assays were tested for diagnostic performance parameters in concordance with the requirements described in the Annex I of the IVDR (EU 2017/746). A mix of commercially available known-truth samples and patient samples previously analyzed at our institute with alternative technologies, like qPCR, pyrosequencing or FISH, were used for this purpose.

### 2.5. Statistical Analyses

Statistical analyses have been performed as specified in the respective graphs (Mann–Whitney test or Kruskal–Wallis test, as appropriate). Numerical data are reported as medians (range), and categorical data are reported as absolute frequencies (%). *p*-values ≤ 0.05 were considered statistically significant. Statistical analyses were performed with GraphPad Prism, GraphPad Software, San Diego, CA, USA.

## 3. Results

### 3.1. Patient Characteristics

Between 1 January 2019 and 30 September 2023, DNA- and RNA-based NGS was performed in 316 consecutive NSCLC cases with squamous histology. The cohort included 211 (66.8%) men and 105 (33.2%) women. The median age was 68 years (range 36–85). Nearly all patients had a smoking history (97.9% being either current or former smokers) and the median number of pack-years was 40 (range 0–200). Since we performed reflex molecular testing independent of tumor progression, our study cohort included cases in stage I (n = 52, 21.1%), stage II (n = 41, 16.6%), stage III (n = 88, 35.6%) and stage IV (n = 66, 26.7%). DNA- and RNA-based NGS was performed on different types of samples, most commonly on biopsies (n = 255, 80.7%), followed by resection specimens (n = 56, 17.7%) and cytology/cell block specimens (n = 5, 1.6%). Most commonly, lung tissue was the source for molecular testing (n = 271, 85.8%). In the remaining cases, specimens from lymph nodes, distant metastases or pleura were tested. In accordance with current guidelines, PD-L1 testing using the tumor proportion score (TPS) was performed in all cases with a median TPS of 5% (range 0–100%). The main patient characteristics of the cohort are summarized in Table 1.

### 3.2. DNA-Based NGS Results

The median tumor cell content of the examined tissue area was 40% (range 10–90%). Mutations could be detected in all 22 included genes, with varying frequency in different cases (Figure 1, Appendix A). The most commonly mutated gene was *TP53* (n = 228, 72.2%), followed by *PIK3CA* (n = 37, 11.7%) and *PTEN* (n = 20, 6.3%). In 7/316 (2.2%) cases, an *EGFR* mutation could be detected (3× deletion in exon 19, 2× p.L858R in exon 21, 1× duplication in exon 20, and 1× p.R494T in exon 12). A total of 5/6 of the *BRAF*-mutated cases harbored a point mutation in exon 11, with one exception being a case with *BRAF* p.K601E mutation in exon 15. Although no *BRAF* p.V600E mutation was detected, there is evidence that cases with the described *BRAF* p.K601E mutation might also be responsive to tyrosine kinase inhibitors [12,13]. *KRAS* mutations were present in 15/316 (4.7%) cases in our cohort, with 7/316 (2.2%) showing the now targetable *KRAS* G12C mutation. *ALK* mutations were detected in 4/316 (1.3%) cases, all being point mutations, two in exon 23 and two in exon 29. Mutations in *FGFR1*, *FGFR2* or *FGFR3* were present in 17/316 (5.4%) cases. All detected mutations within clinically relevant genes and their exact locations are visualized in Appendix A (lollipop plots).

In addition to mutations, amplification events could be suspected in 23/316 (7.3%) cases due to detected differences in gene coverage. Subsequent confirmatory tests (e.g., CNV analyses) have not been routinely performed. Of note, amplification events are increasingly recognized as clinically relevant in NSCLC and should, thus, be reported if present [14]. In our cohort, they could be observed in *EGFR*, *ERBB2*, *FGFR1*, *FGFR3*, *KRAS*, *MET* or *PIK3CA* (Figure 1 and Appendix A). 

**Figure 1 cancers-16-00903-f001:**
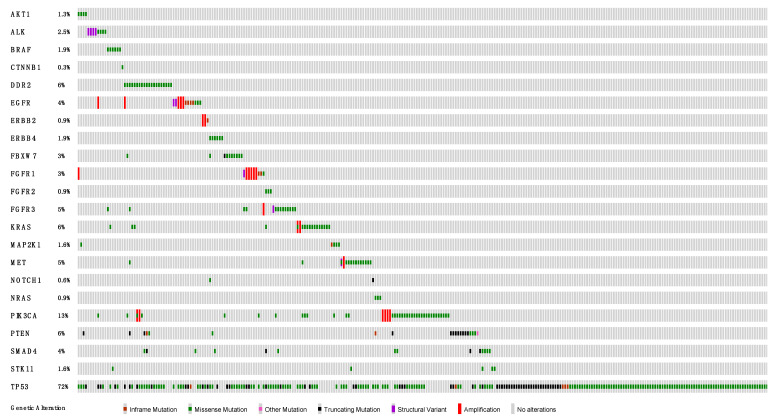
Oncoprint of all detected genetic alterations. Columns correspond cases and rows to genes. Percentages represent the incidence of alterations in respective genes within our cohort. Types of genetic alterations are color-coded and are shown in the legend at the bottom. Analysis and visualization were performed with Oncoprinter from cBioPortal [15,16].

### 3.3. RNA-Based NGS Results

Genetic rearrangements could be detected in 8/316 (2.5%) cases, with the *EML4-ALK* fusion being the most common one (n = 2, 0.6%). Furthermore, single cases harbored a *KRT6A-ALK*, *SLC24A3-ALK*, *MAD1L1-EGFR*, *EGFR-NUP160*, *ASH2L-FGFR1* or *FGFR3-TACC3* fusion, respectively. The detected translocation events of all eight cases are visualized in Figure 2. Because in the so far undescribed *SLC24A3-ALK* fusion, the 3′ partner gene maps to an intron, we additionally performed ALK immunohistochemistry, yielding a strong positive result and thus demonstrating its functional relevance. In addition to fusion events, a *MET* exon-14 skipping alteration could be detected in one case with the help of RNA-based NGS.

### 3.4. Detection of Already Established Therapeutic Targets in NSCLC

A plethora of genetic alterations are already well-established therapeutic targets for an approved drug in NSCLC, recently summarized in [14,17]. In our cohort of NSCLC with squamous histology, 21/316 (6.6%) cases harbored one of these alterations. Clinicopathological characteristics of these cases are summarized in Table 2. Among these 21 cases were seven with a *KRAS* G12C mutation, six with an *EGFR* mutation, four with an *ALK* fusion, two with an *EGFR* fusion, one with a *BRAF* p.K601E mutation, and one with a METex14 skipping event. In addition to the detected molecular drug targets, co-mutations were observed in 13 cases, most commonly within *TP53* (11/21 cases, 52.4%), and in single cases also within *CDKN2A*, *KEAP1*, *MET*, *PIK3CA*, *SMAD4*, and *STK11*. Of note, 16/21 (76.2%) patients harboring a molecular drug target were female. Appendix A summarizes the response evaluation criteria in solid tumors (RECIST) for all cases harboring an established molecular target and receiving targeted therapy.

Because most NSCLC guidelines recommend molecular testing in SCC only in never/light smokers (≤15 pack-years) or young (<50 years) patients [6,7], we correlated our molecular findings with these clinical characteristics. Of note, the majority of our patients harboring a molecular drug target had >15 pack-years (Figure 3A), and all were >50 years of age (Figure 3B), undermining restrictive molecular testing strategies. Furthermore, molecular biomarkers could be detected across all stages (Figure 3C).

Recently, it has been questioned whether molecular targets typically occurring in AC can also be found in SCC or whether these target-bearing SCCs in fact represent undersampled adenosquamous carcinomas (ASC), histologically a mixture of AC and SCC components that per definition can only be diagnosed on resection specimens but not on biopsies [3,18]. To address this issue, we searched for subsequent resection specimens of these patients that harbored targetable alterations and compared the subsequent diagnosis with that of the primary sample where molecular testing had been performed. Paired biopsy and resection specimens were available in ten of these cases. In only 3/10 (30%) cases, the definite diagnosis was then ASC, while the majority of cases (7/10, 70%) represented pure SCC, proving that therapeutic targets do also occur in this histological subtype.

### 3.5. Detection of Emerging Therapeutic Targets in NSCLC

In addition to already well-established therapeutic targets, there is an increasing number of genetic alterations regarded as emerging biomarkers for targeted therapy in NSCLC [14]. At least one of these emerging biomarkers recently listed by Hofman et al. was present in 87/316 (27.5%) cases of our cohort. The most common ones included genetic alterations of *PIK3CA* (44 cases) and *FGFR1-3* (26 cases). Similar to established biomarkers, most patients harboring an emerging biomarker had >15 pack-years and were >50 years of age (Figure 4A,B). Emerging molecular biomarkers could be detected across all stages (Figure 4C). Furthermore, the presence of an emerging target was significantly associated with a lower stage, while the presence of an established target was irrespective of stage (Appendix A), substantiating the importance of a stage-agnostic molecular testing workflow.

### 3.6. Correlation of PD-L1 Status with Established and Emerging Molecular Biomarkers

Apart from oncogenic mutations, personalized treatment decisions are largely influenced by PD-L1 expression and there are several clinical trials examining the benefit of combining targeted therapies with immune checkpoint inhibitors in lung SCC (NCT03735628; NCT04721223; NCT04000529). Therefore, we examined the association between PD-L1 expression and detected genetic alterations within potentially targetable oncogenes in our cohort (Appendix A). Interestingly, genetic alterations within FGFR3 were associated with a significantly lower PD-L1 TPS, possibly influencing future treatment considerations.

## 4. Discussion

NSCLC is a success story of biomarker-driven targeted therapy approaches. Today, most guidelines agree that broad molecular reflex testing should be performed in all Acs [8]. In contrast, testing in SCCs is more controversial. While some guidelines recommend molecular profiling only in patients that meet certain clinical criteria (e.g., non-smoker, young age), others recommend it in all SCC patients, irrespective of smoking status or age [6,7,8]. To add real-world evidence to this controversy, we performed a single-center study examining the value of broad molecular reflex testing in an unselected cohort of 316 consecutive lung SCC cases. 

From a technical point of view, molecular testing in NSCLC becomes increasingly challenging, with ever more genetic alterations to be covered and the most common testing material being small biopsies with often low numbers of tumor cells. Thus, adequate tissue management that guarantees an economical use of biopsy specimens is key. One of the most important measures in this regard is to avoid additional leveling steps in the tissue block because this would unnecessarily use up much of the tissue. Thus, unstained slides might be cut as soon as histomorphology points towards a diagnosis of NSCLC. These slides can then be stored until molecular testing is ordered by the clinician (clinician-initiated molecular testing) or immediately further processed by the molecular pathology laboratory (pathologist-initiated molecular testing, “reflex testing”). Of note, pathologist-initiated reflex testing in NSCLC has decidedly been recommended by a recent expert consensus paper [17]. 

Importantly, our NGS-based reflex testing approach for NSCLC is irrespective of clinical parameters, including tumor stage and smoking status. Hence, our cohort included a wide range of tumor stages and is therefore relevant both for palliative and neoadjuvant/adjuvant treatment considerations. The importance of this stage-agnostic molecular profiling approach is reflected by the recently published results of the ADAURA trial that led to the approval of osimertinib in the adjuvant setting of resectable stage IB-IIIA NSCLCs harboring specific EGFR mutations [19]. Substantiating this, in our cohort, both established and emerging targets could be detected across all stages, with emerging targets even being significantly associated with lower stages. Furthermore, our cohort consisted of an exceedingly high proportion of current or former smokers, reflecting the generally high smoking rate in Austria and other European countries [20]. Thus, the often-proclaimed recommendation to perform molecular profiling in SCC only in non-/light smokers would have deprived many of our patients of the chance to potentially receive biomarker-driven targeted therapy. Furthermore, the recommended pathologist-initiated reflex testing workflow is hardly feasible when the (often unreliable) smoking status has to be waited out, with the consequence of unnecessarily losing precious tissue and time.

Due to the high molecular complexity of lung SCC, combination therapies might be essential for better treatment efficacy. Combining targeted therapy with immunotherapy seems particularly promising [21]. Therefore, the relationship between specific tumor mutations and immune-related biomarkers might be of interest for future treatment considerations. We have observed that genetic alterations in FGFR3 are associated with a significantly lower PD-L1 TPS. This is in line with a recent study that identified the E3 ubiquitin ligase NEDD4 as a key factor for PD-L1 degradation in FGFR3-activated cancer, providing a mechanistic link for potential drug combinations in the future [22].

Although our study is one of the first real-world studies specifically dedicated to lung SCC, there are other NSCLC studies that have also included SCC cases. For example, Griesinger et al. reported on biomarker testing in 3717 advanced NSCLC cases, 796 of them being of squamous histology. Within the latter subgroup, 4.4% harbored a genetic alteration in EGFR, 1.4% in BRAF, 1.2% in ROS1, and 0.5% in ALK, testing results that are comparable with our experience [23]. The cohort by Adib et al. included 337/3115 (10.8%) SCC cases, showing that 5% of them harbor targetable alterations [24], again being largely consistent with our findings. In a study using DNA-based NGS in a cohort of 172 lung SCCs, Sands et al. identified potentially targetable mutations in 38% of cases, the most common being PIK3CA mutations [25]. In a recent paper from the nNGM network in Germany examining MET aberrations in NSCLC, Kron et al. reported four cases with MET ex 14 skipping mutations in stage IIIB/IV SCC, further arguing in favor of testing all NSCLC cases, irrespective of histological subtype [26].

The limitations of our study are its retrospective nature and the lack of long-term clinical outcomes, e.g., survival data. However, we were able to include the response evaluation criteria in solid tumors (RECIST) for all cases harboring an established molecular target and receiving targeted therapy. In addition, by including all consecutive SCC cases tested by DNA- and RNA-based NGS within a specified timeframe, we could exclude a selection bias. Furthermore, real-world studies like ours have the advantage of reflecting the routine setting, thus being easily applicable in future clinical practice.

In accordance with other real-world molecular testing studies that included lung SCC cases, the frequency of detected therapeutic targets in SCC is lower than in AC [9,23,24]. Although SCC patients harboring a targetable mutation have a better outcome when treated with a TKI, this effect might not be as pronounced as in AC [27]. These aspects might raise financial considerations because the reimbursement of molecular testing costs is different across countries and sometimes a challenge even for AC [28]. A mixture of AC and SCC components, with each component comprising ≥10% of the tumor, is defined as adenosquamous carcinoma. These tumors might have a mutation profile similar to AC and are therefore promising candidates for targeted therapy. Importantly, the definitive diagnosis of adenosquamous carcinoma requires a resection specimen. However, the most common specimen for primary histopathological evaluation is a small biopsy that often includes only one of the components, either AC or SCC [3]. Underscoring this problem, adenosquamous carcinomas masquerading as pure SCC have been reported [29], further substantiating the need to not exclude NSCLC cases with squamous histology from reflex molecular testing approaches.

By including RNA-based NGS in our reflex testing approach, we were able to detect gene rearrangements in eight cases. Intriguingly, five of these fusion events have never been described in the literature so far: *KRT6A-ALK*, *SLC24A3-ALK MAD1L1-EGFR*, *EGFR-NUP160* and *ASH2L-FGFR1*. Although our assay for RNA-based NGS has been thoroughly validated (see Section 2), these rare fusion events and their clinical relevance need to be confirmed in future studies. In recent years, *EGFR* fusions in particular have been in the spotlight with several case reports/series being reported in lung AC. Although there is currently no approved drug specifically for *EGFR* fusions, many of these cases have been treated with EGFR TKIs and antitumor responses have been documented, both in vitro and in clinics [30,31,32,33]. Due to prevailing testing strategies that are often restricted to lung AC, data about fusion events in lung SCC are exceedingly rare [34]. However, based on our findings of potentially relevant fusion events, this might be a promising field for future clinical studies.

## 5. Conclusions

In conclusion, we provide real-world evidence that DNA- and RNA-based reflex testing in NSCLC with squamous histology is of clinical relevance. More specifically, we demonstrate that the majority of cases harboring already established or emerging therapeutic targets were >50 years of age and had >15 pack-years, patient groups that have so far often been excluded from molecular testing strategies. Therefore, we propose a reflex testing workflow using DNA- and RNA-based NGS for all patients with NSCLC, irrespective of histology, but also irrespective of age or smoking status (Figure 5). So far, molecular testing in NSCLC with squamous histology is only rarely implemented in centers across Europe [35]; however, we are convinced that the success story of biomarker-driven personalized medicine in NSCLC can only be further continued with a non-selective broad molecular profiling approach.

## Figures and Tables

**Figure 2 cancers-16-00903-f002:**
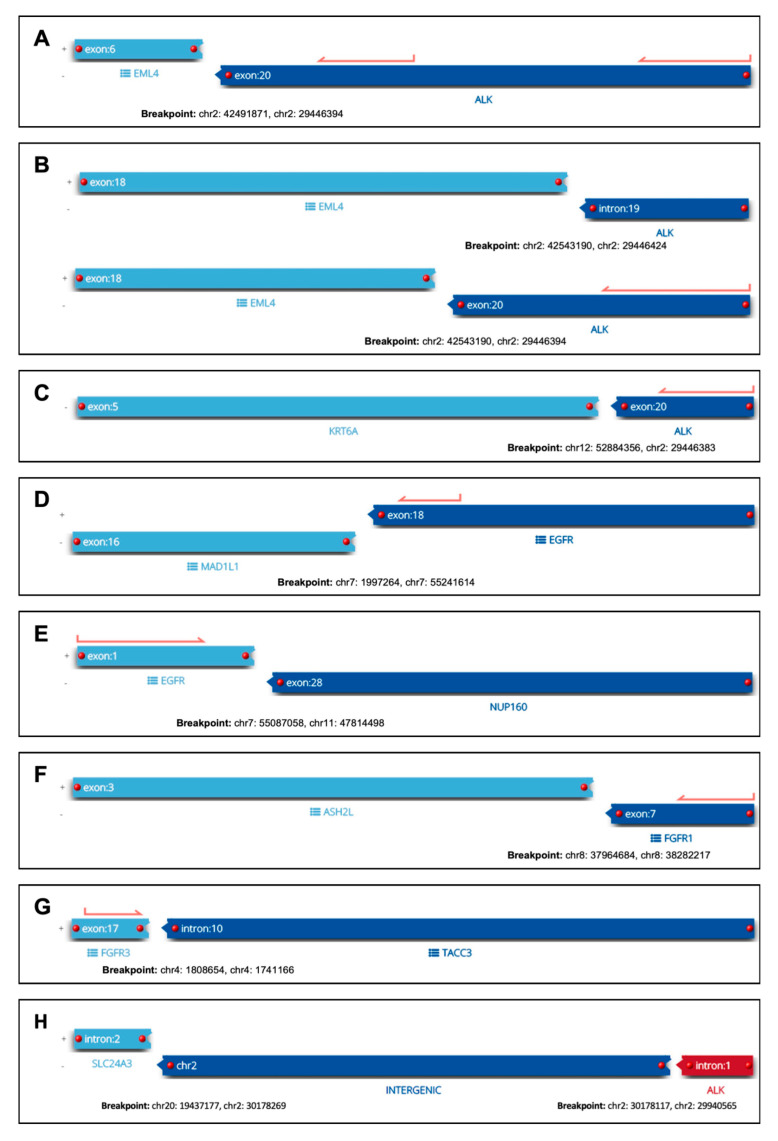
Visualization of detected fusions. Panels (**A**–**H**) represent the seven cases with translocation events. For each case, the involved genes, exons/introns and genomic locations are shown.

**Figure 3 cancers-16-00903-f003:**
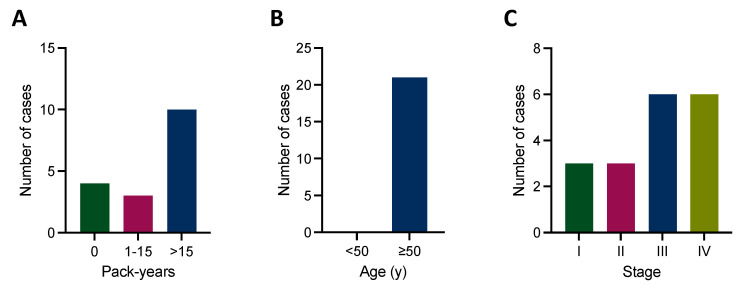
Established therapeutic targets and clinical characteristics. (**A**) The majority of cases harboring an established therapeutic target have >15 pack-years, and (**B**) all cases harboring an established therapeutic target are >50 years of age. (**C**) Established therapeutic targets could be detected across all tumor stages.

**Figure 4 cancers-16-00903-f004:**
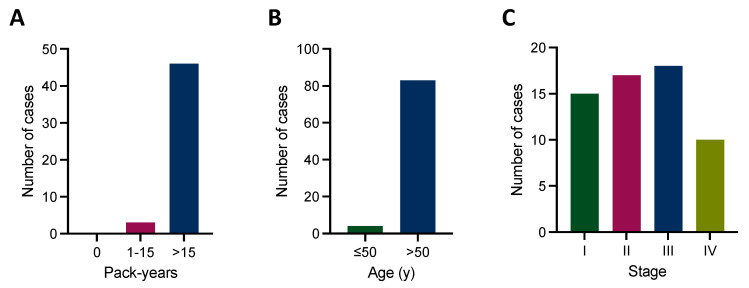
Emerging therapeutic targets and clinical characteristics. (**A**) The majority of cases harboring an emerging therapeutic target have >15 pack-years, and (**B**) are >50 years of age. (**C**) Emerging therapeutic targets could be detected across all tumor stages.

**Figure 5 cancers-16-00903-f005:**
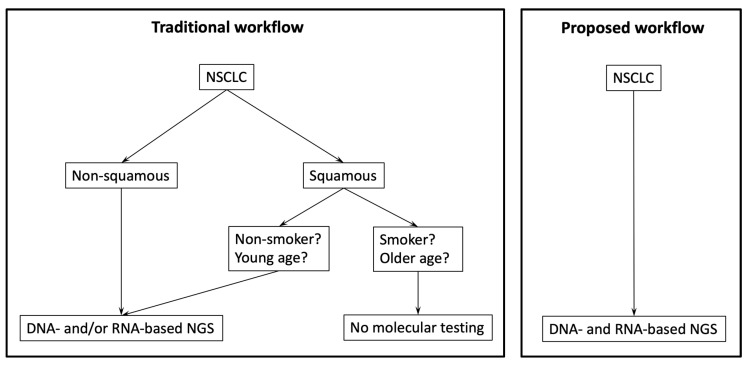
Comparison of routine molecular workflows for NSCLC. The currently most often used workflow is shown on the left, the proposed workflow based on data presented in this article is shown on the right.

**Table 1 cancers-16-00903-t001:** Patient characteristics of the study cohort.

Basic Characteristics	n = 316
Age—median (range)	68 (36–85)
Female	105 (33.2%)
Male	211 (66.8%)
**Smoking status known**	**n = 242**
Smoker	237 (97.9%)
Current smoker	151 (63.7%)
Former smoker	86 (36.3%)
Never smoker	5 (2.1%)
**Pack-years known**	**n = 201**
≤15 pack-years	15 (7.5%)
>15 pack-years	186 (92.5%)
Pack-years median (range)	40 (0–200)
**Staging known**	**n = 247**
Stage I	52 (21.1%)
Stage II	41 (16.6%)
Stage III	88 (35.6%)
Stage IV	66 (26.7%)
**Sample type**	**n = 316**
Resection specimen	56 (17.7%)
Biopsy	255 (80.7%)
Cytology/Cell block	5 (1.6%)
**Sample origin**	**n = 316**
Lung	271 (85.8%)
Pleura	2 (0.6%)
Lymph node	24 (7.6%)
Distant metastasis	19 (6%)

**Table 2 cancers-16-00903-t002:** Clinicopathological characteristics of cases harboring molecular targets for already approved drugs.

Case	Sex	Age	Smoker	PY	Stage	Molecular Target	Co-Mutations	PD-L1 (TPS)	Tested Specimen (Site/Type)	Subsequent Sample	Subsequent Diagnosis
1	f	72	yes (former)	40	IIB	*EML4-ALK* fusion	no	5	lung biopsy	lung resection	SCC
2	f	61	no	0	IVA	*EML4-ALK* fusion	no	0	lung biopsy	adrenal resection	ASC
3	f	74	yes (former)	30	IVA	*KRT6A-ALK* fusion	no	0	lung biopsy	no	NA
4	f	75	yes (former)	30	IIA	*SLC24A3-ALK* fusion	no	10	lung biopsy	lung resection	SCC
5	m	76	yes (current)	50	IB	*EGFR-NUP160* fusion	*TP53* p.H179R	10	lung biopsy	lung resection	SCC
6	f	69	no	0	IIA	*MAD1L1-EGFR* fusion	no	0	lung biopsy	lung resection	ASC
7	f	70	yes (former)	15	IVB	*EGFR* p.S768_D770dup	*TP53* p.E298X	10	brain resection	no	NA
8	f	70	NA	NA	IIIB	*EGFR* p.L747_P753delinsS	no	1	lung biopsy	lung resection	ASC
9	f	60	yes (former)	8	IIIA	*EGFR* p.746_750del	*TP53* p.177_182del	60	lung biopsy	no	NA
10	f	67	yes (current)	20	IVB	*EGFR* p.L747_P753delinsS	*TP53* p.M246fs	90	lung biopsy	no	NA
11	f	81	no (current)	0	IIIA	*EGFR* p.L858R	*SMAD4* p.W524X	20	lung biopsy	lung resection	SCC
12	f	69	no	0	IB	*EGFR* p.L858R *EGFR* p.S768I	*TP53* p.R213Q	2	lung biopsy	lung resection	SCC
13	m	61	NA	NA	NA	*BRAF* p.K601E	*CDKN2A* p.25_31del *STK11* p.A398V	NA	lung biopsy	no	NA
14	f	55	NA	NA	NA	*KRAS* p.G12C	TP53 p.V157F	NA	soft tissue biopsy	no	NA
15	f	67	yes (former)	10	IIIB	*KRAS* p.G12C	*PIK3CA* p.H1047R *TP53* V225fs	1	lung biopsy	no	NA
16	m	61	NA	NA	NA	*KRAS* p.G12C	*TP53* p.V274F	70	adrenal biopsy	no	NA
17	m	70	yes (current)	50	IA3	*KRAS* p.G12C	*TP53* p.R213L	100	lung biopsy	lung resection	SCC
18	f	63	yes (current)	45	IVB	*KRAS* p.G12C	no	90	soft tissue biopsy	no	NA
19	f	66	yes (current)	35	IVB	*KRAS* p.G12C	*TP53* p.P190fs *KEAP1* p.Q217X	0	lung biopsy	no	NA
20	f	63	yes (current)	30	IIIA	*KRAS* p.G12C	no	10	lung biopsy	lung resection	SCC
21	m	73	yes (current)	50	IIIA	METex14 skipping	*TP53* p.P190L *MET* p.T1010I	70	lung biopsy	no	NA

Abbreviations: PY, pack-years; TPS, tumor proportion score; NA, not applicable; SCC, squamous cell carcinoma; ASC, adenosquamous carcinoma.

## Data Availability

The authors declare that all the data supporting the findings of this study are available from the corresponding author upon reasonable request.

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
