# Peer review of "Expanding Broad Molecular Reflex Testing in Non-Small Cell Lung Cancer to Squamous Histology"

_cancers, 2024, doi:10.3390/cancers16050903_

Round 1
Reviewer 1 Report (New Reviewer)
Comments and Suggestions for Authors
The study by Zacharias et al. proposes a reflex testing workflow utilizing DNA and RNA NGS approaches to identify all relevant biomarkers in NSCLC, encompassing newly diagnosed cases regardless of histology, age, and smoking status.
While the work is well-executed, there are some areas for improvement based on the following comments:
- At the moment, EGFR fusions are not a molecular target with an approved drug in lung cancer. Please modify the text accordingly.
- I suggest discussing the cost implications in the paper's discussion section. Given that only 6% of SCC cases may benefit from targeted therapy through NGS, and even then may not experience the same level of response as adenocarcinomas, this raises significant financial considerations. Securing reimbursement or financing molecular test for adenocarcinomas diagnosis, both in early and advanced stages, is already challenging, and this aspect warrants attention.
Comments on the Quality of English LanguageMinor editing of English language required
Author Response
Comment 1: - At the moment, EGFR fusions are not a molecular target with an approved drug in lung cancer. Please modify the text accordingly.
Response 1: We thank the reviewer for this important comment. We agree that there is currently no approved drug specifically for EGFR fusions. However, several case series have convincingly documented significant antitumor responses of lung cancer cases harboring an EGFR fusion treated with already approved EGFR TKIs (e.g., Konduri et al. EGFR Fusions as Novel Therapeutic Targets in Lung Cancer. Cancer Discovery 2016). To emphasize the reviewer’s important point in the revised manuscript, we have modified the respective section in the text (see lines 346-350).
Comment 2: I suggest discussing the cost implications in the paper's discussion section. Given that only 6% of SCC cases may benefit from targeted therapy through NGS, and even then may not experience the same level of response as adenocarcinomas, this raises significant financial considerations. Securing reimbursement or financing molecular test for adenocarcinomas diagnosis, both in early and advanced stages, is already challenging, and this aspect warrants attention.
Response 2: We thank the reviewer for this good suggestion. In the revised manuscript, we have now complemented the discussion on this important topic (see lines 331-334). Furthermore, we have added a reference that describes the situation (including financial aspects) of molecular testing in lung cancer across different countries (Thunnissen et al, TLCR 2020). These additions will provide readers a balanced view on this important topic.
Reviewer 2 Report (New Reviewer)
Comments and Suggestions for Authors
Zacharias et al. described the real world data conducted in their single center. They showed clinically meaningful genetic alteration in certain proportion of patients of NSCLC-SCC. Although some studies showed the similar results, their study convinced previous studies and give the useful evidence in this field. I appreciate this paper for the clinical management of NSCLC SCC.
Author Response
Comment: Zacharias et al. described the real world data conducted in their single center. They showed clinically meaningful genetic alteration in certain proportion of patients of NSCLC-SCC. Although some studies showed the similar results, their study convinced previous studies and give the useful evidence in this field. I appreciate this paper for the clinical management of NSCLC SCC.
Response: We thank the reviewer for this this very positive feedback on our manuscript.
Reviewer 3 Report (New Reviewer)
Comments and Suggestions for Authors
The Authors present a real-world study regarding an agnostic approach to NGS-sequencing of squamous cell carcinoma of the lung. Even if retrospective this paper could pave the way to sequence all the NSCLCs regardless the phenotype. Even if not original their work is one of the first real-world studies on this controversial subject. Moreover the Authors used very simple NGS panels that are avilable in many Pathology Labs. Therefore their message could receive attention from many Colleagues.
The presentation of the data is accurate and the paper is easily readable.
Only some consideration to reinforce the text:
a) All the samples were fixed in formalin and included in paraffin. Was the DNA treated with uracyl DNA glycosilase to remove deaminated cytosines that could have caused a non biological relecant C>T transition?
b) Lines 170-174. " Amplification events could be suspected"...
Gene amplification is gaining importance in all the solid tumors. My question is: have you confirmed your suspects of amplification performing a simple FISH test at least for ERBB", MET, FGFR1 and FGFR3?
Author Response
Comment 1: All the samples were fixed in formalin and included in paraffin. Was the DNA treated with uracyl DNA glycosilase to remove deaminated cytosines that could have caused a non biological relecant C>T transition?
Response 1: We thank the reviewer for this important technical question. In our practice, DNA extracted from paraffin blocks older than 5 years is routinely treated with uracyl DNA glycosilase to reduce deamination artifacts. Due to our reflex testing workflow in NSCLC where we perform molecular testing immediately after diagnosis, this time interval has not been reached in cases from the current study.
Comment 2: Lines 170-174. " Amplification events could be suspected"...
Gene amplification is gaining importance in all the solid tumors. My question is: have you confirmed your suspects of amplification performing a simple FISH test at least for ERBB", MET, FGFR1 and FGFR3?
Response 2: Thank you for this interesting and relevant comment. We completely agree that gene amplifications are gaining importance in solid tumors, including NSCLC. This is also the reason why we always write a comment for the clinician when we suspect an amplification event based on mutation testing that says for example: “All PIK3CA amplicons show an increased coverage in sequencing, suggesting an amplification of the PIK3CA locus. This observation could be verified in a CNV analysis.“ From the cases included in the current study, no subsequent CNV analysis or FISH test has been requested from the clinicians and therefore not been performed. Because the strength of our study is to provide real-world data from clinical practice, we only included analyses that have been performed in the routine setting. In the revised manuscript, we have now clarified the reviewer´s question in the text (see lines 169-170).
This manuscript is a resubmission of an earlier submission. The following is a list of the peer review reports and author responses from that submission.
Round 1
Reviewer 1 Report
Comments and Suggestions for Authors
In the manuscript entitled “Expanding Broad Molecular Reflex Testing in Non-Small Cell Lung Cancer to Squamous Histology”, the authors are reporting their results from DNA/RNA NGS testing on consecutive SCC (N=316), performed in daily routine in their institution. It is a well-written paper on a timely topic, for which I would like to congratulate the authors. Nonetheless, I would recommend some adaptations (minor revisions) in order to further improve the work:
- Abstract: line 23: adenocarcinoma should be substituted by “non-SCC”, as this is to my knowledge the correct recommendation for molecular testing.
- Finding a targeted mutations does not necessarily mean that the patients will respond to the therapy, which was evaluated in clinical studies in a different histological tumor type. I can remember to have read that SCC might respond worse to TKIs, e.g. in the following publication: Joshi A, Zanwar S, Noronha V, et al. EGFR mutation in squamous cell carcinoma of the lung: does it carry the same connotation as in adenocarcinomas? OncoTargets Ther 2017; 10: 1859–1863. It would be beneficial to the claim on “clinical significance” to include a more balanced discussion with this regard.
- Is the frequency of targetable mutations in SCC associated with stage? Please include this.
- I am sure this is not the only manuscript that reports targetable mutations in SCC. The authors could include a small paragraph citing published work and setting this in relation to their findings. Were the frequencies of targetable mutations different to other cohorts? I can recall e.g., one paper from the nNGM network in Germany where a significant amount of MET ex 14 skipping mutations has been reported in stage IIIB/IV SCC (Kron A, Scheffler M, Heydt C, et al. Genetic Heterogeneity of MET-Aberrant NSCLC and Its Impact on the Outcome of Immunotherapy. J Thorac Oncol 2021; 16: 572–82.), also arguing in favor of testing all NSCLC regardless of histological subtype.
- Even though I am personally strongly in favor of reflex testing, the argumentation provided in lines 274-277 is not correct, as unstained slides could still be cut in advance (as currently recommended) also in the non-reflex-testing-workflow and stored until testing is ordered. Thus, no additional cutting of the bloc would be necessary. The authors could rephrase this paragraph to avoid discrediting their point/recommendation by an erroneous argumentation.
Table 2: please provide abbreviations in the table legend (it is not immediately clear what ASC means).
Please correct some minor typos/grammar issues:
- Line 173: delete events or the “s” in amplifications.
- Line 307: “However” should start a new sentence.
- Line 219: I would propose “in the clinics”
Author Response
Response to Reviewer 1
Comment 1: Abstract: line 23: adenocarcinoma should be substituted by “non-SCC”, as this is to my knowledge the correct recommendation for molecular testing.
Response 1: We thank the reviewer for pointing this out. In the revised manuscript, we have changed the wording accordingly (lines 23-24)
Comment 2: Finding a targeted mutations does not necessarily mean that the patients will respond to the therapy, which was evaluated in clinical studies in a different histological tumor type. I can remember to have read that SCC might respond worse to TKIs, e.g. in the following publication: Joshi A, Zanwar S, Noronha V, et al. EGFR mutation in squamous cell carcinoma of the lung: does it carry the same connotation as in adenocarcinomas? OncoTargets Ther2017; 10: 1859–1863. It would be beneficial to the claim on “clinical significance” to include a more balanced discussion with this regard.
Response 2: Thank you for bringing this to our attention. We have now elaborated on this important point in the discussion section (lines 349-350). Furthermore, we have included the reference suggested by the reviewer (Joshi et al, OncoTargets Ther 2017).
Comment 3: Is the frequency of targetable mutations in SCC associated with stage? Please include this.
Response 3: We thank the reviewer for this good suggestion. In the revised manuscript, we have now included this aspect in the results section (see lines 217-218 and 257-261). Furthermore, we included the parameter “stage” in the respective figures (Figures 3C, 4C, and S2). Interestingly, both established and emerging targets could be detected across all stages (Figures 3C and 4C), with emerging targets even being significantly associated with lower stages (Figure S2), providing further evidence for a stage-agnostic testing approach.
Comment 4: I am sure this is not the only manuscript that reports targetable mutations in SCC. The authors could include a small paragraph citing published work and setting this in relation to their findings. Were the frequencies of targetable mutations different to other cohorts? I can recall e.g., one paper from the nNGM network in Germany where a significant amount of MET ex 14 skipping mutations has been reported in stage IIIB/IV SCC (Kron A, Scheffler M, Heydt C, et al. Genetic Heterogeneity of MET-Aberrant NSCLC and Its Impact on the Outcome of Immunotherapy. J Thorac Oncol 2021; 16: 572–82.), also arguing in favor of testing all NSCLC regardless of histological subtype.
Response 4: We thank the reviewer for bringing this to our attention. Accordingly, we have now extended the discussion with an additional paragraph dedicated to other studies reporting targetable mutations in SCC and comparing their results with ours (lines 327-339).
Comment 5: Even though I am personally strongly in favor of reflex testing, the argumentation provided in lines 274-277 is not correct, as unstained slides could still be cut in advance (as currently recommended) also in the non-reflex-testing-workflow and stored until testing is ordered. Thus, no additional cutting of the bloc would be necessary. The authors could rephrase this paragraph to avoid discrediting their point/recommendation by an erroneous argumentation.
Response 5: We thank the reviewer for this valuable comment. We agree that the original argumentation on this point was not correct and that this might have discredited our main claim. Thus, we corrected this part of the discussion accordingly (lines 293-301).
Comment 6: Table 2: please provide abbreviations in the table legend (it is not immediately clear what ASC means).
Response 6: Thank you for bringing this to our attention. We have now added abbreviations in the legend below the table (line 248).
Comment 7: Please correct some minor typos/grammar issues: Line 173: delete events or the “s” in amplifications. Line 307: “However” should start a new sentence. Line 219: I would propose “in the clinics”.
Response 7: Thank you for these suggestions. We have corrected all mentioned minor typos/grammar issues accordingly (line 178, line 355, line 367).
Reviewer 2 Report
Comments and Suggestions for Authors
1.This research focused on Expanding Broad Molecular Reflex Testing in Non-Small Cell Lung Cancer to Squamous Histology , after check the pubmed, there were so many articles aboult this topic, so this manuscript was not very prospective and significant.
2.This manuscript foucus on clinical problems of lung cancer, with strong clinical value and importantce,very interesting research, and also met the submission topic of this journal,the results was real and the conclusion was convincing, but some places can be more perfect.
3. Use DNA and RNA-based NGS to find some gene mutation especially some gene fusion mutation, maybe important for target therapy, but you not support some results that gene mutation or gene fusion mutation related with the target therapy or survival time.
4. You got the conclusion that “Furthermore, our broad RNA-based NGS 34 approach enabled us to detect five gene fusions that have never been described in the literature so far: KRT6A-ALK, SLC24A3-ALK, EGFR-NUP160, MAD1L1-EGFR, and ASH2L-FGFR1. ”But in 2021 year, have found that EGFR-NUP160 fusion(PMID: 35004252).
5. Gene fusion mutation was much more dangerous or influence the target therapy than single gene mutation?
6.Only focus on smoke or age, not consider other parameters?
7.What was difference between Figure 3 and 4, looks have >15 pack-years and >50 years of age were main factor, no any change.
8. References should be renewable such as PMID: 35004252, PMID: 37270698.
9.Language can be more polish.
Comments on the Quality of English Language
nearly ok.
Author Response
Comment 1: This research focused on Expanding Broad Molecular Reflex Testing in Non-Small Cell Lung Cancer to Squamous Histology , after check the pubmed, there were so many articles aboult this topic, so this manuscript was not very prospective and significant.
Response 1: We thank the reviewer for the opportunity to clarify. Although there are many other papers describing molecular testing results in NSCLC, nearly all of them have mainly focused on lung adenocarcinoma. In contrast, our study is one of the first real-world studies specifically focusing on SCC, thus providing clinically relevant novel insights.
Comment 2: This manuscript foucus on clinical problems of lung cancer, with strong clinical value and importantce,very interesting research, and also met the submission topic of this journal,the results was real and the conclusion was convincing, but some places can be more perfect.
Response 2: We thank the reviewer for this very positive feedback. The input from him/her and the other reviewers helped to further improve the manuscript in many aspects (see marked changes throughout the revised manuscript).
Comment 3: Use DNA and RNA-based NGS to find some gene mutation especially some gene fusion mutation, maybe important for target therapy, but you not support some results that gene mutation or gene fusion mutation related with the target therapy or survival time.
Response 3: Thank you for these valuable suggestions. Unfortunately, the evaluation of survival times is not feasible in the current study because the interval from biomarker testing to the study endpoint is too short for proper evaluation. Still, in the revised manuscript we have now added a separate table describing the targeted therapy that patients in our cohort harboring the respective molecular target received. Furthermore, we added the type of treatment response in this table, showing that a considerable number of patients responded to the targeted therapy that they received (see Table S3).
Comment 4: You got the conclusion that “Furthermore, our broad RNA-based NGS 34 approach enabled us to detect five gene fusions that have never been described in the literature so far: KRT6A-ALK, SLC24A3-ALK, EGFR-NUP160, MAD1L1-EGFR, and ASH2L-FGFR1. ”But in 2021 year, have found that EGFR-NUP160 fusion(PMID: 35004252).
Response 4: We thank the reviewer for his/her diligent background research. However, the mentioned study (PMID: 35004252) has been published by ourselves and the mentioned fusion has not been described in any other literature so far.
Comment 5: Gene fusion mutation was much more dangerous or influence the target therapy than single gene mutation?
Response 5: Thank you for the opportunity to clarify. In our cohort, targeted alterations comprised both gene fusions and mutations. Both types of alterations led to appropriate targeted therapies with often beneficial responses of patients (see Table 2 and Table S3).
Comment 6: Only focus on smoke or age, not consider other parameters?
Response 6: We thank the reviewer for this good suggestion. In the revised figures 3 and 4 we have now also included the parameter “stage”, demonstrating that both established and emerging molecular targets could be detected across all tumor stages (see revised figures 3 and 4).
Comment 7: What was difference between Figure 3 and 4, looks have >15 pack-years and >50 years of age were main factor, no any change.
Response 7: Thank you for the opportunity to clarify. Figure 3 shows the clinical characteristics (pack-years, age, stage) of cases with established molecular targets, and figure 4 shows the clinical characteristics of cases with emerging molecular targets, thus these are different patient subgroups.
Comment 8: References should be renewable such as PMID: 35004252, PMID: 37270698.
Response 8: We thank the reviewer for these literature suggestions. The first reference (PMID: 35004252) has already been included in our manuscript. The second reference (PMID: 37270698) is interesting but does not fit into our manuscript; it might be better suited for clinical studies examining the benefit of specific treatments for NSCLC patients.
Comment 9: Language can be more polish.
Response 9: Thank you for this comment. In the revised manuscript, the language has been improved in several parts of the manuscript (see for example line 178, line 355, and line 367).
Reviewer 3 Report
Comments and Suggestions for Authors
My main concern is with the statistical analyses. Although briefly presented in the Methods section (Section 2.5) there are no descriptors in footnotes or Figure legends to indicate what statistical analyses were completed. It is my bias that all of these data are likely to require analyses by nonparametric methods. Also, there is a need for multifactor and interaction inclusion in the statistical modeling. I recommend consultation with a statistician to address my concerns.
Tables 1 and 2 - Were data examined for effects of outcome relateed to sex, sample time and origin?
Figures 1 and 2 - should have been analyzed as I have nidcated for Table 1.
Figure 3 - Should have statistical analysis for Panel A.
Discussion and study limitations - need to be addressed based on the study design, requested statistical analyses (see above) and potential confounding of findings by age, sex and sample type and origin.
Author Response
Comment 1: My main concern is with the statistical analyses. Although briefly presented in the Methods section (Section 2.5) there are no descriptors in footnotes or Figure legends to indicate what statistical analyses were completed. It is my bias that all of these data are likely to require analyses by nonparametric methods. Also, there is a need for multifactor and interaction inclusion in the statistical modeling. I recommend consultation with a statistician to address my concerns.
Response 1: We thank the reviewer for his thoughtful comments. All statistical analyses used are now described in the figure legends - see figure S2 (Mann-Whitney test), and figure S3 (Kruskal-Wallis test). Furthermore, we agree that multifactor / interaction analyses would be appropriate in a clinical study evaluating for example patient outcome. However, our study is purely descriptive as our main goal is to raise awareness that established and emerging molecular targets are also present in lung squamous cell carcinoma cases across different smoking groups (Figure 3A and 4A), age groups (Figure 3B and 4B), and stages (Figure 3C and 4C), as these patient groups have so far often been excluded from molecular testing strategies.
Comment 2: Tables 1 and 2 - Were data examined for effects of outcome relateed to sex, sample time and origin?
Response 2: Thank you for the opportunity to clarify. No specific outcomes were defined or examined because both table 1 and 2 are purely descriptive overviews of the study cohort. Thus, these examinations would apply for clinical outcome studies but not for our purely descriptive cohort study.
Comment 3: Figures 1 and 2 - should have been analyzed as I have nidcated for Table 1.
Response 3: Thank you again for the opportunity to clarify. Figure 1 and 2 are purely descriptive graphical overviews of the detected genetic alterations. The aim of these figures is to provide the reader with the exact type of mutations and fusions, not to examine a specific clinical outcome.
Comment 4: Figure 3 - Should have statistical analysis for Panel A.
Response 4: We thank the reviewer for this suggestion. However, as mentioned above, the goal of this figure is to raise awareness that molecular targets in lung squamous cell carcinoma cases are present across different smoking groups, including heavy smokers. The plot does not show any statistical analysis, as it is a purely descriptive graph showing the number of cases (y-axis) across different pack-year groups (x-axis).
Comment 5: Discussion and study limitations - need to be addressed based on the study design, requested statistical analyses (see above) and potential confounding of findings by age, sex and sample type and origin.
Response 5: We thank the reviewer for this valuable comment. In the revised manuscript, we have now added a paragraph that includes the limitations of our study (lines 340-346).
Reviewer 4 Report
Comments and Suggestions for Authors
The study presented in the manuscript provides valuable insights into the potential benefits of broad molecular reflex testing in NSCLC cases, particularly those with squamous histology. The authors' retrospective analysis of 316 consecutive lung squamous cell carcinoma cases using DNA- and RNA-based NGS revealed previously established molecular targets for approved drugs in a subset of patients.
While the study's findings are promising and suggest the potential benefits of broader molecular reflex testing, several points should be further addressed.
(1) While the study successfully identified novel gene fusions through RNA-based NGS, employing complementary detection methods such as fluorescence in situ hybridization (FISH) or immunofluorescence (IF) to assess the changes in the levels of KRT6A-ALK, SLC24A3-ALK, EGFR-NUP160, MAD1L1-EGFR, and ASH2L-FGFR1 gene fusions would further strengthen the evidence supporting these alterations and their potential clinical relevance.
(2) To strengthen the study's findings and ensure robustness, it is advisable to complement the internal analysis with external validation using publicly available databases. This approach would provide an opportunity to validate the novel gene fusions identified, and compare the results with existing genomic data from diverse patient cohorts.
(3) Finally, the clinical implications and outcomes associated with the identified molecular alterations warrant exploration to assess their impact on treatment response and patient survival. By correlating the identified molecular alterations with patient prognosis, the study can provide deeper insights into the potential impact of these alterations on disease progression and treatment response, thereby increasing the clinical utility of the data.
Author Response
Comment 1: While the study successfully identified novel gene fusions through RNA-based NGS, employing complementary detection methods such as fluorescence in situ hybridization (FISH) or immunofluorescence (IF) to assess the changes in the levels of KRT6A-ALK, SLC24A3-ALK, EGFR-NUP160, MAD1L1-EGFR, and ASH2L-FGFR1 gene fusions would further strengthen the evidence supporting these alterations and their potential clinical relevance.
Response 1: We thank the reviewer for this important comment. While we fully agree that complementary detection methods would further validate our findings, in practice this is currently not feasible for us because we neither use FISH nor IF for fusion detection in our clinical practice. The implementation of FISH and IF protocols for each of the newly described fusion genes would take way too much time and is thus out of scope for the present study. However, we are grateful for your suggestion as we might look forward to a separate manuscript digging deeper into these rare fusion events.
Comment 2: To strengthen the study's findings and ensure robustness, it is advisable to complement the internal analysis with external validation using publicly available databases. This approach would provide an opportunity to validate the novel gene fusions identified, and compare the results with existing genomic data from diverse patient cohorts.
Response 2: Thank you for this good suggestion. The novel gene fusions have not been described yet and are thus not present in publicly available databases. In the revised manuscript, we have now added a paragraph comparing our molecular data with that of other studies and patient cohorts (lines 327-339). Furthermore, the respective references have been added so that the reader can further compare the molecular testing results of different patient cohorts (references 23-26).
Comment 3: Finally, the clinical implications and outcomes associated with the identified molecular alterations warrant exploration to assess their impact on treatment response and patient survival. By correlating the identified molecular alterations with patient prognosis, the study can provide deeper insights into the potential impact of these alterations on disease progression and treatment response, thereby increasing the clinical utility of the data.
Response 3: We thank the reviewer for these valuable comments and fully agree with the importance of associated clinical outcome data. Although the time interval from molecular testing to the study endpoint is too short for long-term survival analyses, in the revised manuscript we have now included the response evaluation criteria in solid tumors (RECIST) for all cases harboring an established molecular target and receiving targeted therapy, demonstrating that some of our patients indeed received targeted therapy and responded to the therapy (table S3).
Round 2
Reviewer 2 Report
Comments and Suggestions for Authors
The authors have dealed with all my concerns,I agree to accept this manuscript.
Reviewer 3 Report
Comments and Suggestions for Authors
None
Reviewer 4 Report
Comments and Suggestions for Authors
The current version of the manuscript falls short in supplementing corresponding experiments due to time constraints and limited experimental conditions. Additionally, I encourage you to meticulously correct any minor methodological errors and make necessary text edits before resubmitting.